# Genome-Wide DNA Methylation Confirms Oral Squamous Cell Carcinomas in Proliferative Verrucous Leukoplakia as a Distinct Oral Cancer Subtype: A Case–Control Study

**DOI:** 10.3390/cancers17020245

**Published:** 2025-01-13

**Authors:** Alex Proaño, Gracia Sarrion-Perez, Leticia Bagan, Jose Bagan

**Affiliations:** 1Medicina Bucal Unit, Stomatology Department, Valencia University, 46010 Valencia, Spain; alexander.proano@uv.es (A.P.); m.gracia.sarrion@uv.es (G.S.-P.); leticia.bagan@uv.es (L.B.); 2Precancer and Oral Cancer Research Group, Valencia University, 46010 Valencia, Spain

**Keywords:** proliferative verrucous leukoplakia, oral squamous cell carcinoma, DNA methylation, differential methylation, prognosis biomarkers

## Abstract

The aggressiveness of proliferative verrucous leukoplakia patients, characterized by multifocal, progressive white patches with high recurrence rates and malignant transformation, suggests the presence of molecular alterations that have not yet been characterized. Furthermore, oral cancers in patients with proliferative verrucous leukoplakia (PVL-OSCC) exhibit different clinical and prognostic outcomes from those seen in conventional oral squamous cell carcinomas. Here, we compare the genome-wide DNA methylation signatures between conventional OSCC and PVL-OSCC in an attempt to understand whether epigenetic dysregulation might explain the favorable prognosis of PVL-OSCC patients. The identification of methylation biomarkers in oral carcinogenesis can be used for monitoring PVL patients, driving early detection screening strategies for oral cancer, and even enabling the development of new epigenetic therapies. The AGL, WRB, and ARL15 genes were identified as potential prognostic biomarkers. The identified differentially methylated genes could help us better understand the molecular carcinogenesis in PVL-OSCC and the potential use of AGL, WRB, or ARL15 as prognosis biomarkers.

## 1. Introduction

Proliferative verrucous leukoplakia (PVL) is a potentially malignant oral disorder (OPMD) with the highest transformation rate into oral cancer [1,2,3]. Recently, in an update coordinated by the World Health Organization Collaborating Centre for Oral Cancer, this disease of unknown origin has been officially recognized for the first time as a distinct form of classical oral leukoplakias (cOLs) characterized by a progressive, persistent, and irreversible clinical course with the presence of multiple oral leukoplakias that frequently become verrucous [1]. In early stages, there are no pathognomonic clinicopathological features that allow differentiation between frictional keratosis, homogeneous leukoplakias, oral submucous fibrosis, and oral lichenoid diseases [4,5]. In advanced stages, lesions progress to multifocality with a wide spectrum of lesion types ranging from smooth or fissured white patches to exophytic, verrucous, and erythematous lesions, as well as erythroleukoplakias [6,7,8]. Finally, during their evolution, it is observed that between 43.9% (95% CI: 31.9–56.1) [3] and 65.8% (95% CI: 55.3–76.2) [2] of patients develop at least one malignant transformation into oral cancer, although some authors still indicate higher malignancy rates [5,6,7].

PVL, unlike classic oral leukoplakia, predominantly affects women at advanced ages, without tobacco or alcohol habits [1,7,9], and primarily occurs in different locations such as the gums, palate, alveolar mucosa, and buccal mucosa [1,6,7]. Moreover, the malignant transformation rate is five times higher [3] with high rates of recurrence [10] and requires challenging management.

Head and neck carcinomas comprise a heterogeneous and aggressive group of malignant neoplasms located in different areas of the upper aerodigestive tract. These carcinomas have different risk factors, molecular characteristics, treatment responses, and exhibit different clinical behaviors [11,12,13]. Oral squamous cell carcinoma (OSCC) is one of the most aggressive types, characterized by poor prognosis and high rates of metastasis and locoregional recurrences [14]. For years, it has been discussed and hypothesized that oral cavity cancers present subtypes with a range of clinical, biological, and evolutionary differences, such as tongue carcinomas, carcinomas associated with the human papillomavirus, or carcinomas derived from potentially malignant oral disorders [1,15,16]. Therefore, it is reasonable to assume that oral squamous cell carcinomas in patients with proliferative verrucous leukoplakia (PVL-OSCC) may represent a differentiated phenotype of oral cavity squamous cell carcinoma, and this phenotype could be explained by differential epigenetics.

The etiology and molecular pathogenesis of the malignant transformation of PVL remain unknown and intriguing fields of investigation. Recent reviews have summarized the molecular alterations described in PVL [17,18]. Weak evidence has been suggested for chromosomal instability such as DNA aneuploidy, loss of heterozygosity at locus 9p21, or specific expression of the Mcm protein in PVL [18]. All these genetic alterations ultimately result in aberrant gene expression. Although some transcriptomic studies have revealed differentially expressed genes in PVL related to immune surveillance [19] or with upregulation of many cancer-associated genes [20], their etiology and pathogenesis have not been fully defined [18]. The complexity of PVL disease and the landscape of genetic alterations are insufficient to explain the pervasive gene expression changes and alterations to cellular function in cancer. A new focus lies on gene expression regulatory events such as epigenetic factors. DNA methylation is one of the most well-studied epigenetic mechanisms, and it plays an essential role in regulating gene expression [21,22]. The main targets are the cytosines that precede a guanine nucleotide (CpGs). Regions with a high frequency of CpG sites are called CpG islands, defined as regions with over 200 bp and a CpG dinucleotide percentage greater than 55% [21]. Using a high-definition microarray approach is used to evaluate differential DNA methylation patterns, covering a representative portion of all human CpG sequences throughout the genome [22].

Previous studies from our [23,24] and other groups [25] have elucidated the implication of aberrant DNA methylation in the pathogenesis of PVL. These studies confirmed the role of epigenetic alterations and the potential of methylation markers in PVL without a history of oral cancer compared to healthy controls and suggested novel OSCC prognosis biomarkers [23]. Recently, an integrative analysis of transcriptomic and methylomic data in patients with PVL-OSCC revealed that 20% of the 133 differentially expressed genes were potentially regulated by DNA methylation [24], suggesting differences in their transformation pathways [24,25].

Although carcinomas in PVL patients present different clinical, histopathological, and evolutionary features compared to conventional carcinomas, differences at the molecular level have hardly been studied. The main objectives were (1) to compare the methylome profile differences between patients with oral squamous cell carcinoma preceded or not by proliferative verrucous leukoplakia in an epigenome-wide association study (EWAS) using the high-coverage Illumina Infinium HumanDNAMethylation 850 BeadChip and (2) to identify potential prognostic biomarkers.

## 2. Materials and Methods

### 2.1. Study Design and Patients

We conducted an EWAS on 18 patients treated at the Stomatology and Maxillofacial Surgery Department of Hospital General Universitario de Valencia. All participants provided written informed consent, and study protocols and procedures were approved in compliance with the Helsinki Declaration by the Ethics Committee for Human Research of the Universitat de València (Ref. H1523722754549) and the Consortium of the General University Hospital of Valencia (30 May 2019).

This case–control study was conducted across two patient groups.

Inclusion criteria. Group 1 comprises patients under follow-up for proliferative verrucous leukoplakia (PVL) who have developed the first malignant transformation into oral squamous cell carcinoma (OSCC-PVL), with no prior oncological medical history. The criteria we followed to categorize patients as PVL were those established by Cerero et al. in 2010 [7]. Group 2 comprises patients diagnosed with OSCC, without a history of prior potentially malignant disorders or, of course, PVL. These OSCC diagnoses were established after a biopsy of the lesion was taken, and the histopathological findings were confirmed. In the PVL group, smoking was not a factor implicated in its etiology [26]. In neither group did we determine human papillomavirus (HPV) since it has been shown that in PVL this virus does not play a role in the etiology [27]. In the case of OSCC not preceded by PVL, as the lesions were not located in the oropharynx but in the anterior zone of the mouth, HPV does not play a significant role there either, unlike cancers in posterior zones such as the oropharynx. Therefore, we do not test for HPV in our cases.

Clinical and pathological TNM stage was classified according to the eighth edition of the American Joint Commission on Cancer, 2017 [28]. For each patient, two representative biopsies, including epithelial and connective tissue from the cancerous area, were obtained between 2019 and 2021 for the histopathological and methylation studies. The latter were immediately frozen at −80 °C until the analysis.

### 2.2. DNA Extraction and Genome-Wide DNA Methylation

Total DNA from fresh frozen tissue was extracted using the column-based DNA extraction method (E.Z.N.A. DNA kit and DNeasy Blood & Tissue Kit; Qiagen, Hilden, Germany) according to the manufacturer’s recommendations. DNA concentration and quality control were quantified by Qubit^®^ 2.0 Fluorometer (Quant-iT PicoGreen dsDNA Assay, Life Technologies, Carslbad, CA, USA). DNA integrity was evaluated by electrophoresis performed in a 1.3% agarose gel.

For methylation profiling, we used the validated Infinium HumanMethylationEPIC BeadChip (850K) array (Illumina Inc., San Diego, CA, USA), which integrates over 862,927 CpG sites across the genome and covers 99% of known genes and 95% of CpG islands [29]. Only samples providing 500 ng of high-quality DNA were used for bisulfite conversion using the EZ-96 DNA Methylation kit (Zymo Research Corp., Tustin, CA, USA) following the manufacturer’s instructions for Infinium assays. A total of 4 μL of the bisulfite-converted DNA was processed following the Illumina HD Methylation Assay Protocol. The Bead Chips were scanned with an Illumina iScan System, and intensity values (.idat files) were generated.

### 2.3. Data Normalization

Raw data (IDATs) were normalized using the minfi R package (v 1.38) and functional normalization. CpG markers present on MethylationEPIC were classified based on their chromosome location, the Infinium chemistry used to integrate the marker (Infinium I, Infinium II), and the feature category gene region as per UCSC annotation (TSS200, TSS1500, 5′UTR, 1st Exon, Body, 3′UTR). Additional criteria included the location of the marker relative to the CpG island (open sea, island, shore, shelf). Probe filtering was performed at three levels. Each beta value in the EPIC array was accompanied by a detection *p*-value that represents the confidence of a given beta value. CpGs with a high *p*-value *p* > 0.01 (1620 CpGs) were removed to exclude probes with low quality and high variability [30]. Probes overlapping with single nucleotide polymorphisms were removed because they can alter methylation levels (2932 CpGs), as well as probes associated with sex chromosomes (19,681 CpGs) to avoid potential bias due to gender. Therefore, after filtering, 842,179 CpGs were considered valid for this study. For each CpG site, a specific β-value was obtained that ranged from 0 (no methylation) to 1 (complete methylation). The difference in mean β-values between the groups was indicated as Δβ.

### 2.4. Statistical Analysis and Bioinformatics

To summarize the characteristics of different groups, descriptive statistical tests were performed. Chi-square tests were used to compare proportions. Student’s *t*-tests and ANOVA tests were applied to compare crude means of continuous variables. For all tests, *p* < 0.05 was considered to indicate statistical significance.

Firstly, the unsupervised exploratory analysis was performed using a principal component analysis (PCA) and a heatmap of clustered observations and variables where green represents a higher methylation gain. Secondly, supervised differential methylation analysis was assessed using two different approaches. One of these approaches includes using a rank-based regression model for each CpG [31]. Differentially methylated probes supported by a Benjamini–Hochberg False Discovery Rate (FDR) correction < 0.05 were considered statistically significant due to the large number of comparisons required. The other approach includes an elastic net penalized logistic regression model, which was adjusted to select the CpGs that are able to discriminate between groups [32]. The penalization factor for the elastic net was selected by taking the highest lambda at one standard error from the minimum (one-standard-error rule) from 500 repetitions of 10-fold cross-validation. Then, the median of the 500 lambda values was used as the final penalization factor. The elastic net alpha parameter (regulating the mix of L1 and L2 penalization) was set at 0.4. CpGs with non-zero coefficients after the penalization were selected and, thus, were considered as relevant for discriminating between both groups. Adjusted *p*-values lower than 0.05 were considered statistically significant. All statistical analyses were performed using R (v 4.2.0), R packages glmnet (v 4.1-4), and Rfit (v 0.24.2).

## 3. Results

### 3.1. Patient Characteristics

The clinicopathological, demographic, and evolutionary characteristics of patients are described in Table 1. The mean age and follow-up were higher in PVL-OSCC patients. We only found significant differences in the clinical form of tumor presentation, being more variable in PVL-OSCC as they presented exophytic, erythroplastic, or mixed lesions. Interestingly, PVL-OSCC showed a tendency towards a more favorable prognosis, characterized by less perineural infiltration, reduced depth of invasion, fewer occurrences of cervical lymph node metastasis, and an earlier TNM stage (Table 1).

### 3.2. The Unsupervised Exploratory Analysis Did Not Allow for the Differentiation Between the Different Oral Cancer Groups

In this first approach, an exploratory principal component analysis did not reveal clear differences between both groups (Figure 1a). As expected, no distinct clusters or outliers were observed among the different patient groups. Next, we performed an unsupervised heatmap using 5000 randomly selected CpGs and performed hierarchical clustering analysis on both observations and variables (Figure 1b). The findings were consistent with the previous analysis, showing no clear distinction between groups. This lack of distinct clusters can be explained by the fact that both groups consist of patients with oral squamous cell carcinomas.

### 3.3. Identification of Differentially Methylated CpG Sites

Two supervised differential methylation analyses were performed. The first analysis includes adjusting a rank-based regression model for each CpG methylation value. After correcting for multiple comparisons using the False Discovery Rate, 14 CpGs corresponding to 10 genes showed statistically significant differences between both groups with an adjusted *p*-value < 0.05 (Table 2). Additionally, the results were also represented in a heatmap with values of these CpGs, and the differences were visually confirmed using the 14 CpGs (Figure 2).

An additional second analysis was performed to define the existence of potential specific methylation differences using an elastic net-penalized logistic regression model. The 500 repetitions of the cross-validation procedure selected a lambda value of 0.95. This statistical analysis determined nine CpGs that provide the highest discrimination power (Table 3 and Figure 3).

We also observed that 16 of the 23 CpGs (70%) were hypomethylated in the PVL-OSCC group.

### 3.4. Identification of Potential DNA Methylation Biomarkers

Interestingly, upon crossing the results of the two supervised analyses, it was observed that there were two genes that were significantly differentially methylated in both, one of them corresponding to the WRB gene and the other to the AGL gene.

Furthermore, we used data from a previous study published by our group [33] describing differential methylation patterns between PVL without antecedent or presence of oral cancer, and homogeneous leukoplakias and healthy controls. An integrative analysis was conducted using the 163 differentially methylated CpGs described in PVL patients (none of which are part of the present study). ARL15 was the only gene differentially methylated in PVL patients with or without malignant transformation when compared to homogeneous leukoplakias, cOSCC, and healthy controls.

## 4. Discussion

The aggressiveness of PVL, characterized by multifocality, high recurrence rates, as well as malignant transformation and tendency to develop second primary tumors, suggests the presence of molecular alterations not yet characterized in PVL patients [10,18]. Here, we compare the genome-wide DNA methylation signatures in fresh frozen tissues between conventional OSCC and PVL-OSCC in an attempt to understand whether epigenetic dysregulation might explain their clinical and evolutionary differences.

First of all, we observed that PVL-OSCC patients had a more favorable prognosis than cOSCC. This was evidenced by two published meta-analyses [34,35], especially with regard to the mortality rate (21.29%, CI: 8.77–36.36) [34], recurrence rate (22%) [35], and the uncommon development of metastases in PVL [35]. In contrast, cOSCC exhibited higher mortality rates [36] and cervical lymph node metastasis (40%) [37]. We aim to comprehend if these distinct clinical phenotypes could be explained by differential methylomes.

Studying epigenetic heterogeneity in cancer evolution is critical because epigenetic dysregulation is a hallmark of cancer that provides insight into the mechanisms of disease initiation and progression [38]. DNA methylation is the most studied deregulated epigenetic mechanism in various types of cancer [39], especially in oral cancer [21,40]. DNA methylation regulates gene expression through different molecular mechanisms, one of which is its ability to prevent the transcription factors from binding and directly influence chromatin folding [39]. To our knowledge, this represents the most extensive methylomic study in oral squamous cell carcinomas developing in PVL patients.

Clinical, socio-demographic, and histopathological covariates have been taken into account to ensure that the epigenetic differences could be attributed to the two different groups of oncology patients (OSCC-PVL vs. cOSCC). In our study, we did not find significant differences in either age or toxic habits between both groups of patients. Thus, we avoided these confounding factors in our results. In fact, according to the available evidence, one of the particularities of PVL is the absence of an etiopathogenic association with tobacco [1,2,7]. Therefore, alterations in the methylation status would not be related to tobacco exposure.

In the first approximation, the unsupervised principal component analysis and the hierarchical clustering of the heatmap did not reveal clear differences in the methylome profile between the two groups of patients. This could be because both groups belonged to oral squamous cell carcinoma patients, and therefore their epigenomes are not as differential. In fact, only global differences would be expected when comparing cancer samples and intra-individual healthy tissues [40], cancer and inter-individual healthy controls [41], or in diseases with mutations in DNA methyltransferase enzymes that would produce a global change in the methylation profile [42]. For example, in the study by Jithesh et al. [40], they compared the methylome of 43 OSCC tissues with their adjacent paired healthy tissues and identified two clearly distinct groups. Similarly, the study by Milutin et al. [41] was conducted on cytological samples in which the methylome clearly differentiated the three groups evaluated (oropharyngeal and oral carcinomas, oral lichenoid disease, and healthy controls). A study by Simo-Riudalbas et al. [42], which was carried out on patients with the rare autosomal recessive disease such as immunodeficiency, centromeric instability, and facial anomalies syndrome who had mutations in the DNMT3B gene, showed a global reduction in DNA methylation across all chromosomes. Furthermore, it is noteworthy that in our PCA, we also did not observe defined clusters in each type of tumor sample, which may further reflect the biological heterogeneity of oral squamous cell carcinomas.

The two parallel supervised bioinformatic analyses were able to discriminate PVL-OSCC from cOSCC. From these, we identified 21 differentially methylated CpGs corresponding to 14 genes that would allow discrimination between groups. Of them, three CpGs had not been assigned to any known gene (namely cg26134913, cg11369761, and cg24104268), and the rest had been associated with genes unrelated to oral cancer. Previous studies [23,24,25] have investigated the implication of DNA methylation alterations in PVL. These studies confirmed the role of epigenetic alterations and the potential of methylation markers in PVL and suggested novel OSCC prognosis biomarkers [23] and differences in their transformation pathways [24,25]. However, one of the major limitations is the limited sample size.

In our literature search, we found consistent previous reports regarding the genetic or epigenetic dysregulation of 12 genes (86%) in the prognosis of various types of cancer, such as cervical cancer (CASP7) [43], (BRD9) [44], (FGD5) [45], hepatocellular carcinomas (NR2E1) [46], cutaneous melanoma (ARL15) [47], lung cancer (NR2E1) [48], (AGL) [49], (ZNF777) [50], (ZNF429) [51], prostate cancer (CYP11A1) [52], bladder cancer (C18orf18) [53], renal cancer (ZNF433) [54], and leiomyosarcomas (DAXX) [55]. Similarly, the involvement of some of these genes in important cellular processes such as apoptosis (CASP7) [43], transcriptional regulation (ZNF77, ZNF433, and DAXX) [56], mitochondrial degradation (PITRM1) [57], or steroid biosynthesis (CYP11A1) [52] has been described. Therefore, this dysregulation could promote cancer-associated cell proliferation.

Regarding the aberrant methylation status of these genes, it has been described that BRD9 promoter methylation is associated with increased overall and progression-free survival in cervical cancer [44]. On the other hand, the NR2E1 gene is silenced by hypermethylation mechanisms in hepatocellular carcinomas [46] and lung carcinomas [48]. Similarly, increased methylation of CYP11A1 has been linked to the development of prostate cancer recurrences [52]. The hypermethylation of C18orf18, also known as LINC00526, has been associated with poor prognosis and survival in bladder cancer [53]. Likewise, the hypermethylation of ZNF433 has been related to the progression of clear cell renal cell carcinoma [54] and the differential methylation and overexpression of DAXX in the progression of leiomyosarcoma [55]. FGD5 is a proangiogenic gene associated with tumor progression, whose aberrant methylation has been detected in cervical cancer carcinogenesis [45]. As can be observed, some of these methylation changes have been correlated with gene expression changes. This dysregulation is consistent with our results on methylation-mediated regulation of tumor gene expression in PVL-OSCC [24].

The differential methylation pattern of PVL-OSCC included the hypomethylation of 18 CpGs corresponding to nine genes. Two differentially methylated probes (cg15977137 and cg24794734) were similarly epigenetically altered in both bioinformatics-supervised analyses. AGL, also known as GDE, participates in glycogen degradation (glycogenolysis). Mutations in AGL have been associated with glycogen storage disease type III in different ethnicities. Its role as a tumor suppressor gene in bladder and lung cancer has also been described [58,59]. Loss of expression has been correlated with poor prognosis, aggressive tumor growth, and high mortality rates in bladder cancer, although its mechanism of action is unknown [59]. It is believed that the overexpression of the enzymes SHMT2 and HAS2 with AGL loss drives increased synthesis of glycine and hyaluronic acid, leading to enhanced glucose absorption and metabolism that promotes rapid tumor proliferation and growth [59]. On the other hand, the WRB gene, which is maternally imprinted, has been shown to exhibit a conserved imprinting pattern with methylated maternal alleles in cases of Down syndrome [60]. Additionally, a significant association between strabismus and the WRB gene polymorphism (rs2244352) has been confirmed in several cohorts [60]. Finally, it has been described that ARL15 overexpression inversely influences circulating adiponectin levels, which has been linked to a higher risk and susceptibility to type 2 diabetes, coronary heart disease, and rheumatoid arthritis [61]. Recently, a bioinformatic study has provided the first evidence that the low expression of ARL15 is associated with a favorable prognosis in cutaneous melanoma [47].

Interestingly, none of the gene promoters commonly hypermethylated in cOSCC, such as p16INK4A, p14ARF, CDKN2A, MGMT, or DAPK, were identified as differentially methylated between our two groups [17,21]. This could be because the methylation levels at these promoters are similar in PVL-OSCC and cOSCC.

Based on these findings, to our knowledge, we are the first to describe the aberrant methylation in these 14 genes as prognosis biomarkers in oral squamous cell carcinomas. This could be explained by the fact that no studies have evaluated this profile in malignant tumor lesions in PVL, the heterogeneity of oral carcinomas, and the use of different platforms in assessing the methylome. The identification of methylation biomarkers in oral carcinogenesis can be used for monitoring OPMD such as PVL, to drive early detection screening strategies for oral cancer, and even enable the development of new epigenetic therapy.

We are aware of the limitations of the present study, due to the limited sample size and the study being underpowered to allow estimating the sensitivity and specificity of the prognostic biomarkers. However, it is important to consider the challenges in obtaining samples from the initial malignant transformation in PVL patients. Studies with more cases are needed to ratify our results, which can be considered preliminary. Still, due to the rarity of this entity (PVL), it is a good starting point to advance in the knowledge of its etiology, which is currently unknown. At the epigenomic level, we have established the molecular basis for future research that focuses on functional characterization of differentially methylated genes, and now we need to validate our results in independent cohorts and multicenter studies in larger cohorts.

## 5. Conclusions

In this case–control study we demonstrate the molecular differences between PVL-OSCC and cOSCC, supporting the hypothesis that molecularly defined subtype classification in oral cancer could improve therapeutic development and proper patient monitoring. Clinical, socio-demographic, and histopathological covariates have been taken into account to ensure that the epigenetic differences are solely attributable to alterations in methylation patterns. As described above, with bibliographic evidence, neither tobacco, alcohol, nor HPV are considered to influence PVL. The identified differentially methylated genes could help us better understand the molecular carcinogenesis in PVL-OSCC and the potential use of AGL, WRB, or ARL15 as prognosis biomarkers.

## Figures and Tables

**Figure 1 cancers-17-00245-f001:**
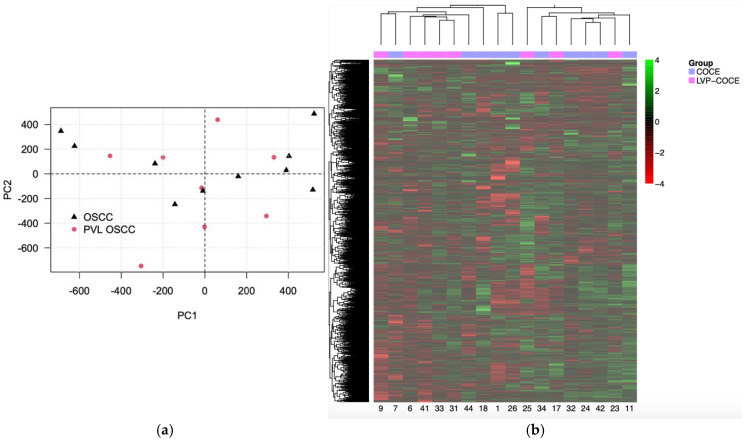
An analysis of the global DNA methylation profile in oral cancers preceded or not by proliferative verrucous leukoplakia. (**a**) Representation of the principal component analysis of the methylation data. (**b**) Heatmap and hierarchical clustering with a random sample of 5000 CpGs. DNA methylation Z-score values were represented as a color scale ranging from green for higher methylation levels to red for lower methylation levels.

**Figure 2 cancers-17-00245-f002:**
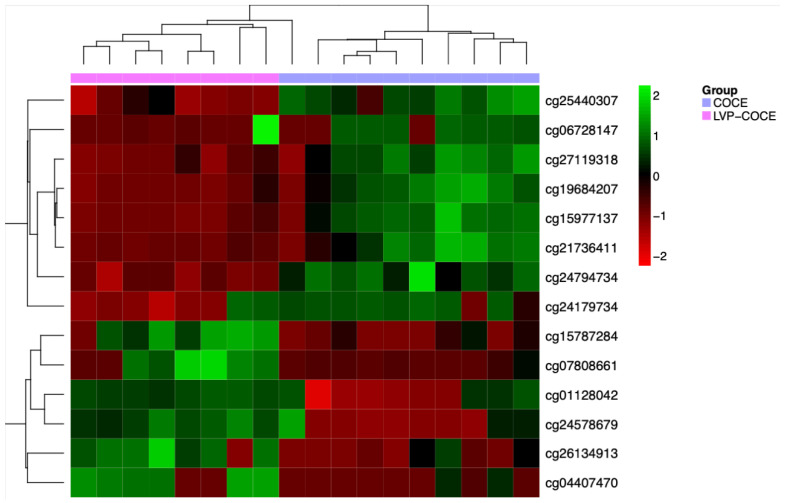
Hierarchical clustering and heatmap with the methylation status of the differentiating CpGs between groups. Rows (CpGs) and columns (patients) are ordered according to the results of a hierarchical clustering algorithm. Z-score color scale ranges from green for higher methylation to red for lower methylation levels.

**Figure 3 cancers-17-00245-f003:**
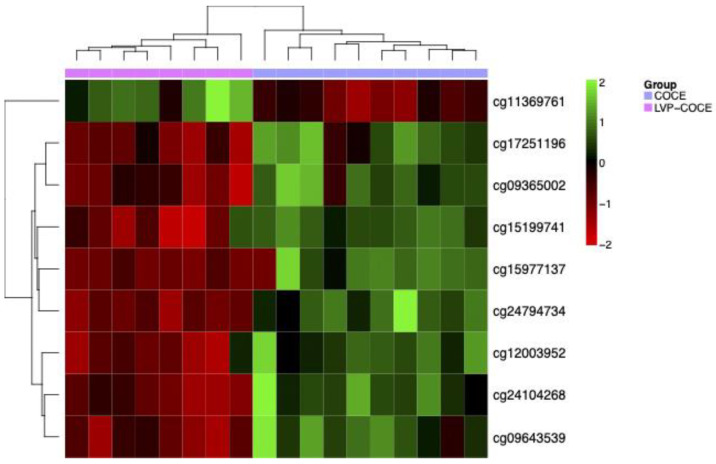
Hierarchical clustering and heatmap of the nine CpGs predicted by the elastic net-penalized logistic regression model.

**Table 1 cancers-17-00245-t001:** Detailed demographic and clinicopathological characteristics of study patients. The patient groups with their clinicopathological and demographic characteristics are the same as those used in the paper ([24]). The present study complements the previous one by evaluating epigenome-wide association differences that were not evaluated in the previous one.

		OSCC-PVL	cOSCC	*p*
**Number of cases (%)**		8	10	
**Mean age ± SD (Years)**		80.25 ± 12.42	72.8 ± 9.05	*p* = 0.062
**Gender (%)**	Female	6 (75)	6 (60)	*p* = 0.502
	Male	2 (25)	4 (40)	
**Tobacco (%)**	Yes	1 (12.5)	4 (40)	*p* = 0.196
	No	7 (87.5)	6 (60)	
**Alcohol (%)**	Yes	0	1 (10)	*p* = 0.357
	No	8 (100)	9 (90)	
**Tumor site (%)**	Gingiva	5 (62.5)	3 (30)	*p* = 0.173
	Palate	2 (25)	1 (10)	
	Floor of the mouth		1 (10)	
	Tongue		4 (40)	
	Buccal mucosa	1 (12.5)		
	Lips		1 (10)	
**Clinical form of the neoplastic lesion (%)**	Erythroplastic	2 (25)		***p* = 0.022**
	Ulceration	1 (12.5)	9 (90)	
	Exophytic	3 (37.5)		
	Mixed	2 (25)	1 (10)	
**Tumor grade (%) ***	G0	2 (25)		*p* = 0.292
	G1	4 (50)	5 (50)	
	G2	1 (12.5)	4 (40)	
	G3	1 (12.5)	1 (10)	
**Cancer infiltration (%)**	Bone	3 (37.5)	3 (30)	*p* = 0.737
	Perineural	2 (25)	7 (70)	*p* = 0.058
	Lymphovascular	2 (25)	2 (20)	*p* = 0.8
**DOI (mm) ****	Median (±SD)	3.62 ± 2.21	6.39 ± 2.18	*p* = 0.075
**Metastasis (%)**	Cervical lymph nodes	1 (12.5)	3 (30)	*p* = 0.375
	Distant metastases	0	1 (10)	
**TNM stage (%) *****	I	1 (12.5)	2 (20)	*p* = 0.212
	II	3 (37.5)	2 (20)	
	III	1 (12.5)		
	IV	3 (37.5)	6 (60)	
**Second primary tumors (%)**	Yes	4 (50)	2 (20)	*p* = 0.18
	No	4 (50)	8 (80)	

* Tumor grade (G0: grade cannot be assessed; undetermined grade. G1: well differentiated; low grade. G2: moderately differentiated; intermediate grade. G3: poorly differentiated; high grade). ** DOI: depth of tumor invasion. *** TNM system: T describes the tumor size, N describes lymph node involvement, and M describes distant metastasis.

**Table 2 cancers-17-00245-t002:** Description of the 14 CpG results of differential methylation analysis by a rank-based regression model.

MethylationEPIC Probe ID	Gene Symbol	Cytoband	∆β	Relation to CpG Island	Regulatory Feature
**cg24794734**	AGL	1p21.2	−0.007	Shore	1st Exon, 5′UTR, TSS1500
**cg24179734**	BRD9	5p15.33	−0.3	Shore	Body
**cg04407470**	NR2E1	6q21	0.49	Island	Body
**cg25440307**	ZNF777	7q36.1	−0.019	Island	Body
**cg06728147**	PITRM1	10p15.2	−0.47	Open sea	Body
**cg01128042**	CASP7	10q25.3	0.35	Open sea	Body
**cg26134913**	-		0.32	Shore	-
**cg24578679**	CYP11A1	15q24.1	0.25	Island	Body, TSS200
**cg07808661**	C18orf18	18p11.31	0.27	Island	Body, TSS200
**cg15787284**	ZNF433	19p13.2	0.28	Island	TSS200
**cg27119318**	WRB/GET1	21q22.2	−0.19	Shore	Body, TSS200
**cg21736411**			−0.08	Shore	Body, TSS200
**cg19684207**			−0.18	Shore	Body, TSS200
**cg15977137**			−0.16	Shore	Body, 1st Exon, 5′UTR

Regulatory features and their relation to CpG islands are annotated according to the Infinium MethylationEPIC v1.0 B4 Manifest. CpG island: region of at least 200 base pairs (bp) with a CG content > 55%. Shore: sequences 2 kb flanking the CpG island. Shelf: sequences 2 kb flanking shore regions. Body: intragenic region. TSS1500: 200–1500 bp upstream of the transcription start site (TSS). TSS200: 0–200 bp upstream of the TSS. Opensea: sequences located outside these regions mean that CpG is not located in a gene and may be located in intragenic areas of information about it that remain unknown. ∆β: the difference in mean methylation levels (β-values) between oral squamous cell carcinoma preceded or not by proliferative verrucous leukoplakia.

**Table 3 cancers-17-00245-t003:** Differentially methylated CpGs from the elastic net-penalized logistic regression model.

MethylationEPIC Probe ID	Gene Symbol	Cytoband	∆β	Relation to CpG Island	Regulatory Feature
**cg24794734**	AGL	1p21.2	−0.007	Shore	1st Exon, 5′UTR, TSS1500
**cg11369761**	-		0.12	Shore	-
**cg15199741**	FGD5	3p25.1	−0.09	Open sea	Body
**cg09643539**	ARL15	5q11.2	−0.14	Open sea	Body
**cg09365002**	DAXX	6p21.32	−0.3	Shore	Body
**cg17251196**			−0.17	Shore	Body
**cg24104268**	-		−0.12	Island	-
**cg12003952**	ZNF429	19p12	−0.07	Shore	TSS1500
**cg15977137**	WRB/GET1	21q22.2	−0.16	Shore	Body, 1st Exon, 5′UTR

## Data Availability

Methylation data have been deposited in the ArrayExpress database at EMBL-EBI (www.ebi.ac.uk/arrayexpress (accessed on February 2023) under accession number E-MTAB-12202.

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
