# Peer review of "Genome-Wide DNA Methylation Confirms Oral Squamous Cell Carcinomas in Proliferative Verrucous Leukoplakia as a Distinct Oral Cancer Subtype: A Case–Control Study"

_cancers, 2025, doi:10.3390/cancers17020245_

Round 1
Reviewer 1 Report
Comments and Suggestions for Authors
hello
thank you for this interesting paper
the title is sound
short objective and abstract are well structured
used key words are sufficient
title and key words are correlated with abstract
clear aim is set in the abstract - nothing to change
2.
in the introduction authors briefly present most important VPL characeristics
all necessary information is presented in a great standard
the vpl occurrence and their troublesome treatment alternatives are very well described
two main author goals are presented at the end of the chan[ter
nothing to improve here, all is sufficient
3
a clear methodology and legal/bioethic standards of the paper are clearly set
authors did divide study group to two groups
pvl was used based on NCCCN guidelines, which is very good, like AJCC
clear methodology is described -nothing to change or re-write in this chapter
the DNA data assessment is described
I'm missing inclusion and exclusion criteria - please improve them - please to characterise the group 1 and group 2 patients - were they alcohol, tobacco or HPV+ related or not? secondly, does a secondary head and neck cancer patients also count in this study group or no?
4
results are well described in separate chapters
presented statistical data are sound
used table 1 - I'm missing abbreviations and explanation on TNM, G0, T, DOI and .... and other used short abbreviations - please do explain them under the table
figure 1 - whats the differences between the green and red color depositions?
table 2 also missing abbreviations under table and detailed table 2 explanation in text
table 2, figure 2 and figure 3 - what is the detailed correlation between them? what is the result?
what was the most common and rarest gene expression in this study? and why?
5
discussion is sufficient
used references are good, especially that there is limited amount of study on such VPL lesions
overall results are presented in good equal modality, nothing to change here
clear limitations are presented
6.
final conclusions are quite ok, but I would add and highlight top 3 most important remarks from this study and also add if tobacco alcohol or HPV+ do influence on VPL occurrence and long term outcomes?
used references are quite ok
thank you for the interesting paper
Reviewer 2 Report
Comments and Suggestions for Authors
Dear authors, your article is very interesting but some parts need to be modified and enriched to merit publication.
In the introduction, the mechanism of the transformation of Proliferative verrucous leukoplakia into cancer needs to be better explained.
Recent reviews have summarised(17,18)..... cite these reviews
In the discussion you have to add and cite studies that have dealt with the same subject and you have to add the limitations to the studies
Reviewer 3 Report
Comments and Suggestions for Authors
Genome-wide DNA methylation confirms oral squamous cell carcinomas in proliferative verrucous leukoplakia as a distinct oral cancer subtype: a case control study
The authors present a comparison of genome-wide DNA methylation profiles between oral squamous cell carcinomas (OSCC) in patients with proliferative verrucous leukoplakia (PVL-OSCC) and conventional OSCC (cOSCC) using the Illumina MethylationEPIC BeadChip platform. Supervised analyses identified two distinct cancer phenotypes and 21 differentially methylated CpGs linked to 14 genes, including potential prognostic biomarkers like AGL, WRB, and ARL15. The findings do a good job of highlighting the importance of epigenetic dysregulation in OSCC, particularly in PVL-associated cases, and provide novel insights, but there are some weaknesses that need correcting. Some suggestions and requests for further details are provided below.
[1] For the simple summary, “proliferative verrucous leucoplakia patients” is not that easy for non-experts to know about. Perhaps “proliferative verrucous leucoplakia patients – an aggressive or persistent white patch in the mucosa of the mouth” or something similar might be better?
[2] Somewhere in the text the authors might want to define CpGs as readers from other oncological disciplines may not know what they are
[3] In the Introduction, may as well be consistent in the number of decimal places reported (sometimes 1, sometimes 2)
[4] The authors mention FDR correction but do not specify whether they used Bonferroni, BH or BY, or what the rationale was for the choice, i.e. whether there was weak or strong dependence in the data structure
[5] Why was logistic regression chosen as the model? XGBoost or LightGBM are better at handling interactions and non-linear relationships.
[6] Table 1, why constrain the table to report only p > 0.05? It can be helpful to know if the value was 0.052 or 0.952 and there is not much use in merging such a range of values under one categorical label.
[7] Also Table 1, and Discussion, how can the reader know whether the markers identified are markers of OSCC-PVL versus cOSCC, or markers of ulcerative carcinomas versus non-ulcerative carcinomas, or markers of simple progression, given that squamous cell carcinoma usually does not ulcerate until it is advanced. This is potentially a weakness of the overall approach, caused by the lack of overall statistical power preventing any analysis of matched types. If the authors do not think this is a weakness or relevant issue, this will need explaining to the non-expert reader. At the very least, this difference between the groups needs to be acknowledged in the Abstract and Discussion.
[8] The manuscript states that the markers are prognostic (or can differentiate between the two carcinoma types) but there is no data in Results that demonstrates this, for example sensitivity and specificity (and their 95% CI) for a prognostic test. With such a small sample size test train splits are not practical but leave one out cross validation would work and allow the presentation of a confusion matrix. If the sample size is too small, and the study underpowered to derive estimates of sensitivity and specificity for such a test, this should be explicitly acknowledged.
[9] The authors are to be commended for following best practices in uploading their data to an appropriate repository
Round 2
Reviewer 3 Report
Comments and Suggestions for Authors
The authors have responded to all of my comments, I wish them good luck with their future research efforts.